# Disease-modifying effects of natural Δ9-tetrahydrocannabinol in endometriosis-associated pain

Alejandra Escudero-Lara[1], Josep Argerich[1], David Cabañero[1†*], Rafael Maldonado[1,2†*]

[1]Laboratory of Neuropharmacology, Department of Experimental and Health Sciences, Universitat Pompeu Fabra, Barcelona, Spain; [2]IMIM (Hospital del Mar Medical Research Institute), Barcelona, Spain

**Abstract** Endometriosis is a chronic painful disease highly prevalent in women that is defined by growth of endometrial tissue outside the uterine cavity and lacks adequate treatment. Medical use of cannabis derivatives is a current hot topic and it is unknown whether phytocannabinoids may modify endometriosis symptoms and development. Here we evaluate the effects of repeated exposure to Δ9-tetrahydrocannabinol (THC) in a mouse model of surgically-induced endometriosis. In this model, female mice develop mechanical hypersensitivity in the caudal abdomen, mild anxiety-like behavior and substantial memory deficits associated with the presence of extrauterine endometrial cysts. Interestingly, daily treatments with THC (2 mg/kg) alleviate mechanical hypersensitivity and pain unpleasantness, modify uterine innervation and restore cognitive function without altering the anxiogenic phenotype. Strikingly, THC also inhibits the development of endometrial cysts. These data highlight the interest of scheduled clinical trials designed to investigate possible benefits of THC for women with endometriosis.

*For correspondence:
david.cabanero@upf.edu (DCñ);
rafael.maldonado@upf.edu (RM)

†These authors contributed equally to this work

Competing interests: The authors declare that no competing interests exist.

## Introduction

Endometriosis is a chronic inflammatory disease that affects 1 in 10 women of childbearing age (*Zondervan et al., 2019*). It is characterized by the growth of endometrium in extrauterine locations, chronic pain in the pelvis and the lower abdomen, infertility, emotional distress and loss of working ability (*Fourquet et al., 2011*; *Márki et al., 2017*; *Zondervan et al., 2019*). Current clinical management provides unsatisfactory outcomes. On the one hand, hormonal therapy has unwanted effects including contraception and emotional disturbances (*Ross and Kaiser, 2017*; *Skovlund et al., 2016*), whereas surgical excision of the growths is associated with high-recurrence rates and postsurgical pain (*Garry, 2004*). Hence, clinical treatments are limited and women often unsatisfactorily self-manage their pain (*Armour et al., 2019*). In this context, marijuana legalization for medical purposes in American and European states has led to increased availability of phytocannabinoids (*Abuhasira et al., 2018*). While cannabis may provide pain relief in certain conditions (*Campbell et al., 2001*), it is unclear whether it may modify endometriosis symptoms or development.

Δ9-tetrahydrocannabinol (THC) is the main psychoactive constituent of the *Cannabis sativa* plant, and multiple animal and clinical studies suggest its efficacy relieving chronic pain (*De Vry et al., 2004*; *Harris et al., 2016*; *King et al., 2017*; *Ueberall et al., 2019*; *Williams et al., 2008*), although controversial results have been obtained in human clinical trials (*Stockings et al., 2018*). However, THC has important side effects including cognitive deficits and anxiety (*Célérier et al., 2006*; *Kasten et al., 2017*; *Puighermanal et al., 2013*). This work investigates the effects of natural THC in a mouse model of endometriosis that reproduces the ectopic endometrial growths and some of the

**eLife digest** Endometriosis is a common disease in women caused by tissue that lines the uterus growing outside the uterine cavity on to other organs in the pelvis. This can cause a variety of symptoms including chronic pelvic pain, infertility, and pain during menstruation or sexual intercourse. These symptoms may contribute to anxiety, depression, loss of working ability and a reduced quality of life.

Currently available treatments for endometriosis, including hormonal therapy and surgery, have a limited effect and can produce unwanted side effects. For example, women who undergo surgery to remove the growths may experience post-surgical pain or a recurrence. As a result, women with endometriosis often rely on self-management strategies like dietary changes or exercise. Although cannabis consumption has a large number of potential side effects and can lead to substance abuse, it has been shown to provide pain relief in some conditions. But it is unknown whether it could be useful for treating endometriosis.

Now, Escudero-Lara et al. have created a mouse model that mimics some of the conditions of human endometriosis: pelvic pain, anxiety and memory impairments. The mice were treated with moderate doses of Δ9-tetrahydrocannabinol (THC), which is the main pain-relieving component of cannabis. The THC reduced pelvic pain and cognitive impairments in the mice with the endometriosis-like condition, but it had no effect on their anxious behavior. Escudero-Lara et al. also noticed that endometrial growths were also smaller in the treated mice indicating that THC may also inhibit endometriosis development.

These experiments suggest that THC may be a useful treatment for patients with endometriosis. Clinical trials are already ongoing to test whether these findings translate to patients with the condition. Although THC and cannabis are readily available in some areas, Escudero-Lara et al. discourage using unregulated cannabis products due to the potential risks.

behavioral alterations of clinical endometriosis. Our data show that THC is effective inhibiting hypersensitivity in the caudal abdominal area without inducing tolerance, as well as reducing the pain unpleasantness associated with endometriosis. Notably, THC also prevents the cognitive impairment observed in mice with ectopic endometrium without modifying anxiety-like behavior at this particular dose. Interestingly, THC shows efficacy limiting the development of ectopic endometrium, revealing disease-modifying effects of this natural cannabinoid.

## Results and discussion

### Ectopic endometrium leads to pain sensitivity in the caudal abdomen, anxiety-like behavior and memory impairment

Our first aim was to characterize a novel experimental procedure to evaluate at the same time nociceptive, cognitive and emotional manifestations of endometriosis pain in female mice. Mice were subjected to a surgical implantation of endometrial tissue in the peritoneal wall of the abdominal compartment or to a sham procedure. Mice receiving ectopic endometrial implants developed persistent mechanical hypersensitivity in the caudal abdominal area, whereas sham mice recovered their baseline sensitivity and showed significant differences in comparison to endometriosis mice since the second week of implantation (*Figure 1a* and *Figure 1—figure supplement 1*). To test whether mechanical hypersensitivity of endometriosis mice was specific to this abdominal region, nociceptive responses were also measured in the hind paw. In this distant area, mechanical sensitivity remained unaltered, indicating that pain sensitization did not generalize to other sites (*Figure 1b* and *Figure 1—figure supplement 2*). To discern whether increased nociception was accompanied by a component of negative affect, a measure of pain unpleasantness was taken on day 14 after the surgeries (*Figure 1c*). Endometriosis mice showed increased nocifensive behaviors to mechanical stimuli when compared with sham mice. Similarly, endometriosis mice exhibited enhanced anxiety-like behavior reflected in lower percentages of time and entries to the open arms of the elevated plus maze (*Figure 1d*). Total arm entries were similar in both groups (*Figure 1d*). In line with these findings, previous rodent models of endometriosis found increased mechanosensitivity in the lower

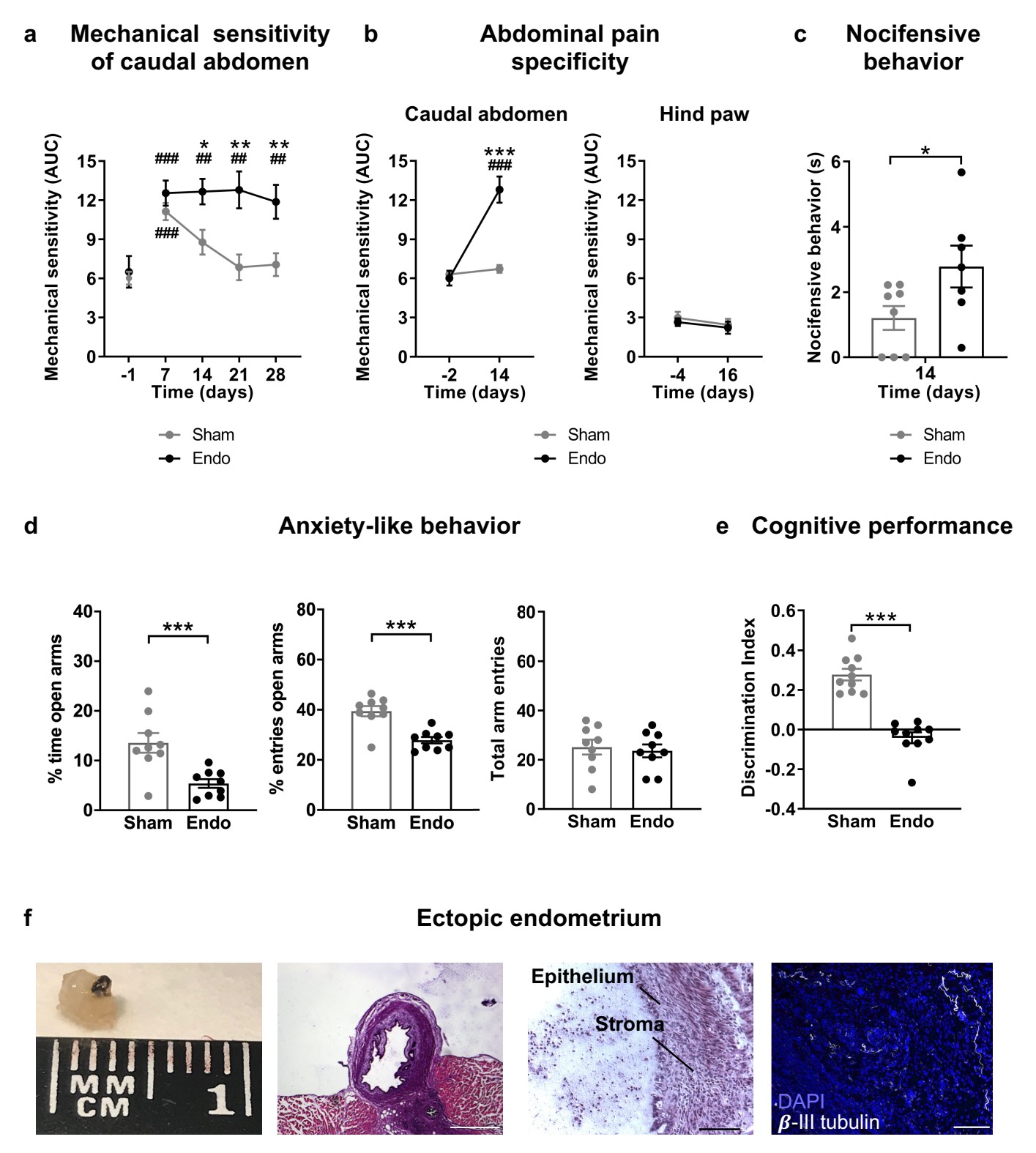

**Figure 1.** Behavioral and histological alterations in female mice with ectopic endometrial implants. Endometriosis mice showed (**a**) persistent mechanical abdominal hypersensitivity that (**b**) was localized in the caudal abdominal area but not detectable in distant areas (hind paw). Mechanical sensitivity is represented by the area under the curve of frequency of response to von Frey filaments. Higher values mean higher mechanical pain. Mice receiving endometrial implants also showed (**c**) increased nocifensive behavior, (**d**) anxiety-like behavior in the elevated plus maze test and (**e**) cognitive

*Figure 1 continued on next page*

*Figure 1 continued*

impairment in the novel object recognition task. (**f**) From left to right: cysts were recovered from endometriosis mice, were filled with fluid (scale bar = 1 mm), contained endometrial epithelium and stroma (scale bar = 100 µm) and were innervated by beta-III tubulin-labeled fibers (scale bar = 100 µm, blue is DAPI and white is β-III tubulin). Error bars are mean ± SEM. One-way repeated measures ANOVA + Bonferroni (**a and b**) and Student t-test (**c, d and e**). *$p<0.05$, **$p<0.01$, ***$p<0.001$ vs sham. ##$p<0.01$, ###$p<0.001$ vs baseline. Endo, endometriosis, AUC, area under the curve.

The online version of this article includes the following source data and figure supplement(s) for figure 1:

**Source data 1.** Effects of ectopic endometrium.
**Figure supplement 1.** Nociceptive responses to abdominal mechanical stimulation with von Frey filaments.
**Figure supplement 2.** Nociceptive responses to abdominal and paw mechanical stimulation with von Frey filaments.
**Figure supplement 3.** Density of beta-III tubulin-labeled fibers in uteri of endometriosis and sham mice.

abdomen (*Arosh et al., 2015*; *Greaves et al., 2017*) and affective-like disturbances (*Filho et al., 2019*; *Li et al., 2018*). Previous works associate nociceptive and emotional distress in chronic pain settings with cognitive decline (*Bushnell et al., 2015*; *La Porta et al., 2015*; *You et al., 2018*), although this cognitive impairment has not yet been revealed in rodent models of endometriosis. We found in our model a dramatic impairment of long-term memory in endometriosis mice (*Figure 1e*). While mnemonic effects of this pathology have not been thoroughly evaluated, a cognitive impairment may contribute to the loss of working ability consistently reported in women with endometriosis (*Hansen et al., 2013*; *Sperschneider et al., 2019*). Hence, mice with ectopic endometrium recapitulate in our model some of the symptomatology observed in the clinics, although manifestations of spontaneous pain could not be evaluated in this work.

Mice receiving endometrial implants developed 3 to 5 endometrial cysts in the peritoneal wall of the abdominal compartment. Cysts were of 2.59 ± 0.34 mm diameter, filled with fluid, with glandular epithelium and stroma and innervated by beta-III tubulin positive fibers (*Figure 1f*), as shown in women (*Tokushige et al., 2006*; *Wang et al., 2009*) and other rodent models (*Arosh et al., 2015*; *Berkley et al., 2004*). Interestingly, we also found increased expression of the neuronal marker beta-III tubulin in the uteri of endometriosis mice (*Figure 1—figure supplement 3*), mimicking not only some of the symptoms but also the histological phenotype observed in women with endometriosis (*Miller and Fraser, 2015*; *Tokushige et al., 2006*).

## Δ9-tetrahydrocannabinol alleviates pain in the caudal abdomen, restores cognitive function and limits the growth of ectopic endometrium

Our second objective was to assess the effects of THC exposure on the endometriosis model to select an appropriate dose for a chronic treatment. Acute doses of THC were first tested in endometriosis and sham mice at a time point in which endometriotic lesions and hypersensitivity in the caudal abdomen were fully developed. Acute THC administration produced a dose-dependent reduction of abdominal mechanical hypersensitivity (*Figure 2*). The acute ED50 of THC 1.916 mg/kg ($\approx 2$ mg/kg) was chosen for the repeated administration.

Repeated exposure to THC 2 mg/kg, once daily for 28 days, provided a sustained alleviation of mechanical hypersensitivity during the whole treatment period (*Figure 3a* and *Figure 3—figure supplement 1*). Repeated THC starting on day 1 could have exerted a preventive effect at endometriosis stages in which pain sensitivity may have not been fully developed. To discern whether the absence in loss of efficacy was due to an inhibition of endometriosis development or to an actual lack of tolerance, we assessed the persistence of THC efficacy once pain was already present. THC given for the first time on day 14 was as effective as THC given on the same day after a daily treatment starting on day 8 (7 days long, *Figure 3b* and *Figure 3—figure supplement 2*). Therefore, THC did not lose its efficacy when repeated administration started once painful symptomatology was established. The absence of tolerance to THC-induced antinociception is in contrast with the tolerance described at higher THC doses in other pain models (*Greene et al., 2018*; *LaFleur et al., 2018*; *Wakley et al., 2014*). As expected, no effects of endometriosis or THC treatments were found in mechanical sensitivity of distant areas (Hind paw, *Figure 3—figure supplement 3*). Endometriosis mice treated with vehicle showed an increase in nocifensive behaviors compared with sham mice (*Figure 3c*). Interestingly, the 7 day treatment with THC inhibited this component of negative affect, while the effects of an acute administration of THC were highly variable. This variable

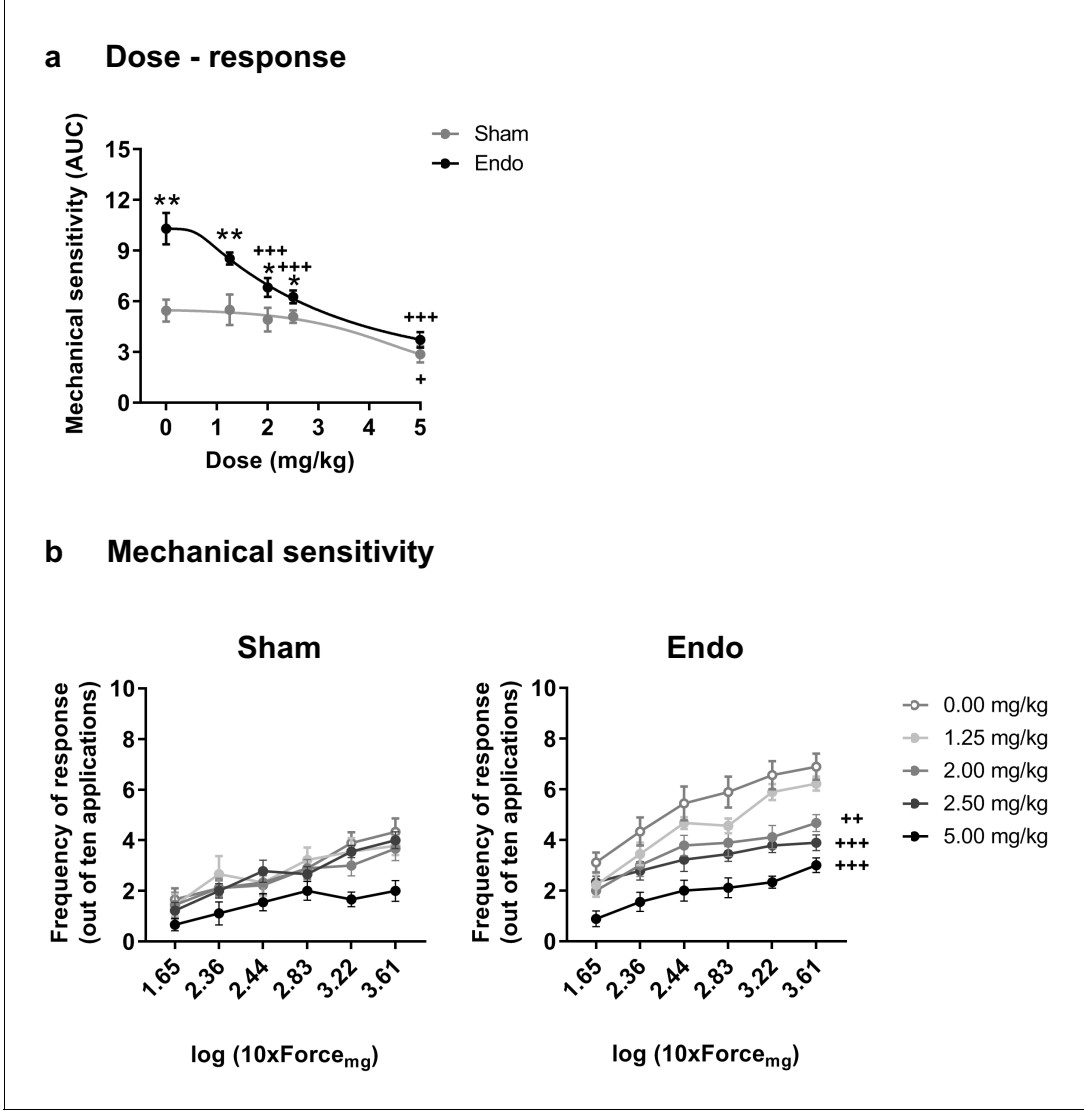

**Figure 2.** Effect of acute THC administration on the nociceptive responses to mechanical stimulation. (a) Acute THC produced a dose-dependent reduction of mechanical hypersensitivity in the caudal abdominal area. Mechanical sensitivity is represented by the area under the curve of frequency of response to von Frey filaments. Higher values mean higher mechanical pain. (b) Administration of 2, 2.5 and 5 mg/kg of THC decreased the frequency of response to von Frey filaments in endometriosis mice. Error bars are mean ± SEM. One-way repeated measures ANOVA + Bonferroni. *p<0.05, **p<0.01 vs sham; +p<0.05, ++p<0.01, +++p<0.001 vs vehicle. Endo, endometriosis; THC, Δ9-tetrahydrocannabinol, AUC, area under the curve. The online version of this article includes the following source data for figure 2:

**Source data 1.** Acute THC effects.

response could be associated to aversive effects associated with a first exposure to THC, an event described in humans (*MacCallum and Russo, 2018*) and mice (*Kubilius et al., 2018*).

Additional experiments were conducted to assess the effects of THC on the anxiety-like behavior induced by endometriosis pain (*Figure 3d*). As in previous experiments, endometriosis mice showed a lower percentage of time in the open arms of the elevated plus maze (*Figure 3d*), revealing increased anxiety-like behavior. However, the percentage of entries to open arms was similar in endometriosis and sham mice. Therefore, the anxiogenic-like effect of ectopic endometrium in these experimental conditions was mild and the present model was not optimal to reveal the emotional component of this painful situation. No significant effects of repeated THC 2 mg/kg were observed on the percentages of time and entries, although THC-treated mice showed a subtle increase in anxiety-like behavior (*Figure 3d*, percentage of time in open arms). Previous studies described

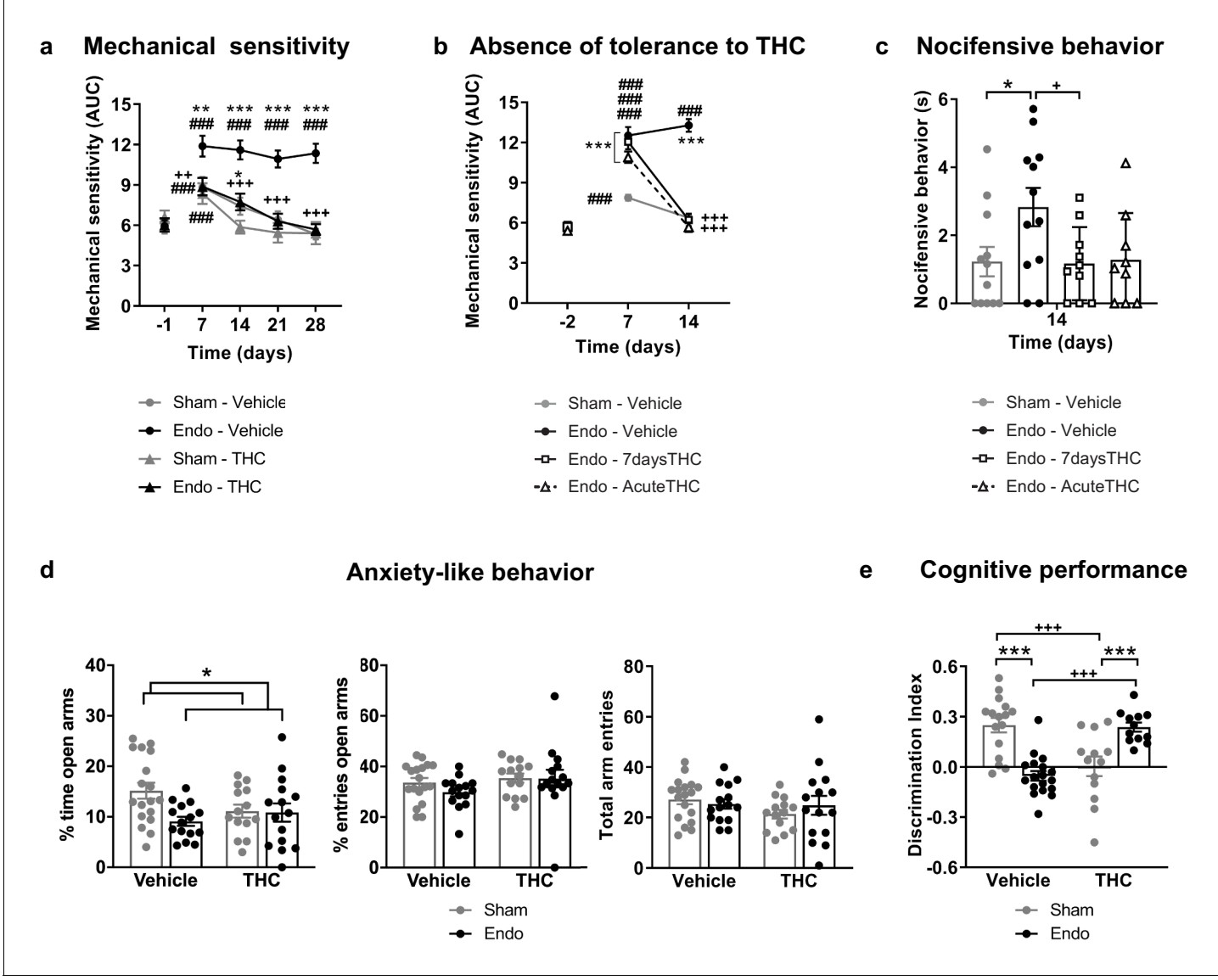

**Figure 3.** Effects of THC on the behavioral changes observed in mice with ectopic endometrium. (a) Repeated THC (28 days) alleviated mechanical hypersensitivity in the caudal abdominal area of endometriosis mice in the von Frey test. (b) THC administered on day 14 after a 6 day treatment (Endo – 7daysTHC) was as effective as an acute dose given on day 14 (Endo – AcuteTHC). Mechanical sensitivity is represented by the area under the curve of frequency of response to von Frey filaments. Higher values mean higher mechanical pain. (c) Nocifensive behaviors were abolished in endometriosis mice after a 7 day treatment with THC (Endo – 7daysTHC). (d) Endometriosis-associated anxiety-like behavior was unaltered after THC in the elevated plus maze test. (e) THC impaired object recognition memory in sham mice and prevented memory deficits of endometriosis mice in the novel object recognition test. THC dose: 2 mg/kg/day. Error bars are mean ± SEM. Two-way repeated measures ANOVA + Bonferroni (a), Mixed model + Bonferroni (b), Kruskal-Wallis + Mann Whitney U (c) and Two-way ANOVA + Bonferroni (d and e). ###$p<0.001$ vs baseline. *$p<0.05$, **$p<0.01$, ***$p<0.001$ vs sham. ++$p<0.01$, +++$p<0.001$ vs vehicle,. Endo, endometriosis; THC, Δ9-tetrahydrocannabinol; AUC, area under the curve.
The online version of this article includes the following source data and figure supplement(s) for figure 3:

**Source data 1.** Effects of repeated THC on behavioral alterations.
**Figure supplement 1.** Effect of chronic THC treatment on nociceptive responses to abdominal mechanical stimulation with von Frey filaments.
**Figure supplement 2.** Effect of a repeated THC treatment starting on day 8 after surgeries on nociceptive responses to abdominal mechanical stimulation with von Frey filaments.
**Figure supplement 3.** Effect of a repeated THC treatment starting on day eight after surgeries on nociceptive responses to hind paw mechanical stimulation with von Frey filaments.

anxiogenic-like effects of slightly higher doses (3 mg/kg) in naïve male mice (*Viñals et al., 2015*), and anxiolytic-like effects when using lower doses (0.3 mg/kg, *Puighermanal et al., 2013*; *Viñals et al., 2015*). Thus, possible effects of THC alleviating pain-related anxiety-like behavior in endometriosis mice could be hindered by intrinsic anxiogenic effects of this THC dose. Therefore, doses with less pain-relieving efficacy could potentially be effective promoting anxiolytic-like effects considering the intrinsic effects of THC on emotional-like behavior. Alternatively, the absence of clear effects of THC on anxiety-like behavior may be associated to the evaluation time point, which was 6 hr after administration to study the impact of pain relief on anxiety-like behavior, rather than to assess direct drug effects. Total arm entries were similar among groups (*Figure 3d*). Memory performance was also assessed the third week after starting the THC treatment. As expected, mice exposed to the chronic nociceptive manifestations of endometriosis showed a pronounced cognitive impairment, as well as sham mice exposed to THC, in accordance with previous reports in naïve males (*Kasten et al., 2017*; *Puighermanal et al., 2013*). Surprisingly, endometriosis mice repeatedly treated with natural THC showed intact discrimination indices (*Figure 3e*) suggesting protective effects of THC in this chronic inflammatory condition. In agreement, recent studies have shown cognitive improvements after THC exposure in old male and female mice (*Bilkei-Gorzo et al., 2017*; *Sarne et al., 2018*).

Exogenous and endogenous cannabinoids have shown modulatory effects on the female reproductive system (*Walker et al., 2019*). Thus, we analyzed the effects of THC on the ectopic and eutopic endometrium and on ovarian follicle maturation. Interestingly, endometriosis mice receiving THC 2 mg/kg for 32 days showed an evident inhibition of the development of endometrial cysts (cyst diameter and area of endometrial tissue, *Figure 4a*) without significant effects on cyst innervation (*Figure 4—figure supplement 1a*). In agreement, a previous study showed antiproliferative effects of WIN 55212–2, a synthetic cannabinoid agonist, on endometrial cell cultures and in ectopic endometrium implanted in immunodepressed mice (*Leconte et al., 2010*). The assessment of the uterine diameter and the area of eutopic endometrium (*Figure 4—figure supplement 1b*) showed no effects of the THC treatment, suggesting that the antiproliferative activity of THC on endometrial cells is restricted to ectopic sites. However, possible effects of THC on established endometriosis lesions were not evaluated. Repeated THC increased the expression of neuronal markers in the uteri of sham mice, similar to the increase provoked by the ectopic endometrium (*Figure 4b*). Interestingly, THC prevented this increase in endometriosis mice (*Figure 4b*) indicating again that THC exposure may have different consequences under chronic inflammatory conditions. In agreement, recent studies showed differential effects of THC on the nervous system of rodents with and without chronic inflammation (*Bilkei-Gorzo et al., 2017*; *Sarne et al., 2018*). To investigate a possible estrogenic influence on these histological findings, we analyzed 17 β-estradiol plasma levels. As expected, 17 β-estradiol plasma levels depended on the phase of the estrous cycle: mice in proestrus had the highest concentration followed by mice in diestrus, and mice in estrus showed the lowest levels (*Figure 4c*, left graph). We found that 17 β-estradiol was similar in all experimental groups (*Figure 4c*, right graph), although the levels of this estrogen were positively correlated with cyst diameter (*Figure 4d*, left), proving the estrogenic influence on ectopic endometrial lesions. 17 β-estradiol levels were not correlated with endometrial area of the cysts (*Figure 4d*, middle), or uterine innervation (*Figure 4d*, right), suggesting independent THC effects on these histological changes.

We also assessed possible effects of THC on ovarian functioning, since previous works have suggested inhibitory effects of THC on folliculogenesis and ovulation (*Adashi et al., 1983*; *El-Talatini et al., 2009*). Numbers of preantral follicles, antral follicles and corpora lutea were similar in all groups in our experimental conditions (*Figure 4—figure supplement 1c*). These data suggest that endometriosis and THC were void of overt effects on ovarian follicle maturation and luteinization, however, other effects of endometriosis or THC on fertility cannot be excluded in our model. Similarly, the presence of prominent symptoms of endometriosis such as dysmenorrhea or dyspareunia could not be evaluated.

## Conclusions

Here we show for the first time that chronic administration of a moderate dose of the phytocannabinoid THC relieves mechanical hypersensitivity of caudal abdominal area, pain unpleasantness and cognitive impairment associated with the presence of ectopic endometrial cysts. These behavioral manifestations correlate with a decrease in the size of ectopic endometrium in THC-exposed mice.

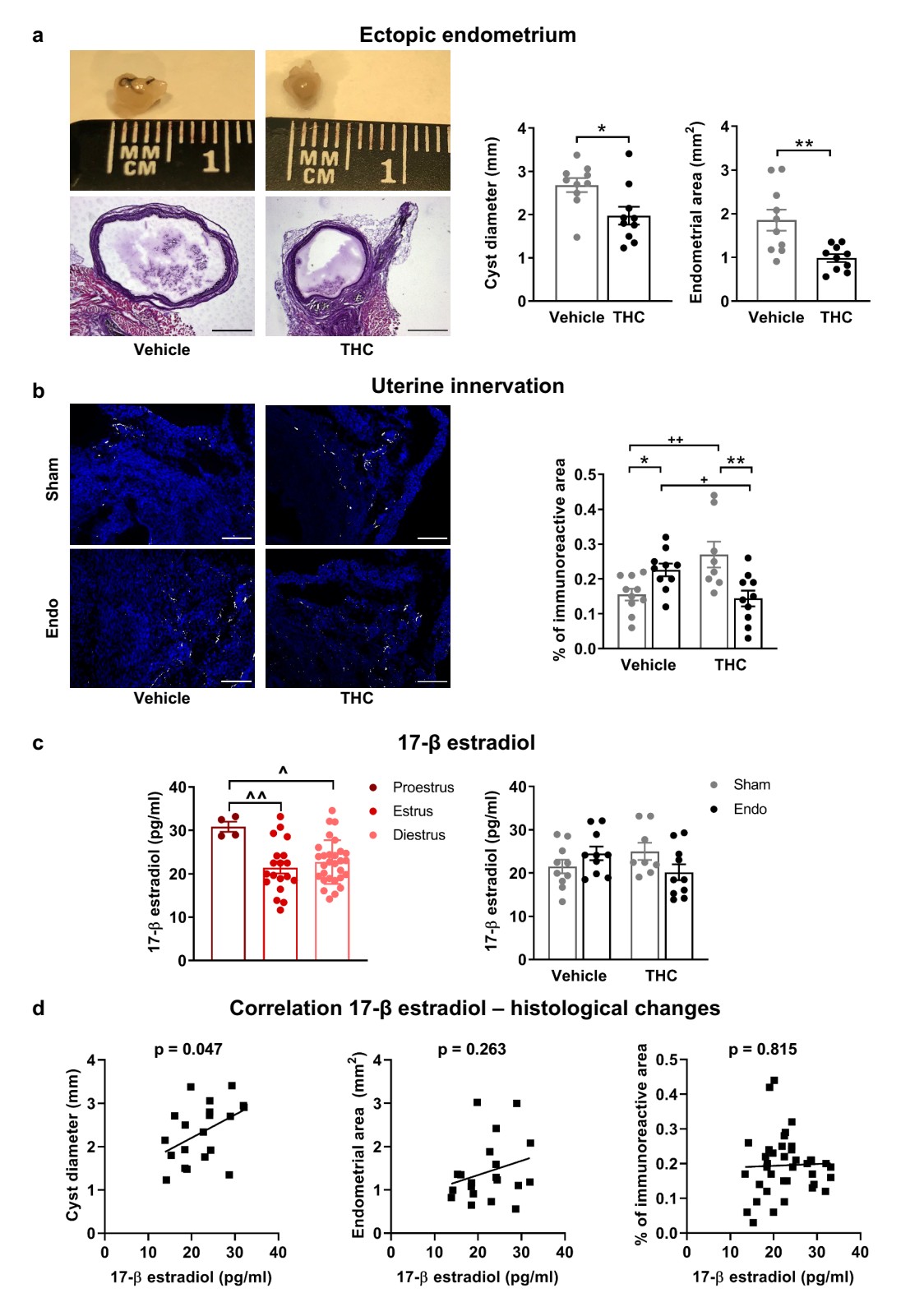

**Figure 4.** Effects of THC on the histological changes observed in mice with ectopic endometrium. (a) Ectopic endometrial growths of mice treated with THC were smaller (left graph) and had less endometrial tissue (right graph) than those of mice receiving vehicle. Scale bar = 1 mm. (b) THC increased innervation in sham mice but prevented uterine hyperinnervation in endometriosis mice. Blue is DAPI and white is β-III tubulin. Scale bar = 100 μm. (c) As expected, 17-β estradiol levels were higher in mice in proestrus (left). Estrogen levels were similar in all experimental conditions (right). (d) There was

*Figure 4 continued on next page*

*Figure 4 continued*

a positive correlation between cyst diameter and plasma levels of 17-β estradiol (left, r = 0.450). Absence of correlation of estrogen levels with cyst endometrial area (middle, r = 0.263) and uterine innervation (right, r = 0.039). THC dose: 2 mg/kg/day. Error bars are mean ± SEM. Student t-test (a, left graph), Mann Whitney U (a, right graph), two-way ANOVA + Bonferroni (b), mixed model + Bonferroni (c, left); Two-way ANOVA (c, right) and Pearson correlation (d). *p<0.05, **p<0.01 vs sham. +p<0.05, ++p<0.01 vs vehicle. ˆp<0.05, ˆp<0.01 vs proestrus. Endo, endometriosis; THC, Δ9-tetrahydrocannabinol.

The online version of this article includes the following source data and figure supplement(s) for figure 4:

**Source data 1.** Effects of repeated THC on histopathological features.
**Figure supplement 1.** Histological features of reproductive tissues after chronic THC treatment.

However, the pain-relieving effects of this particular dose of THC were not accompanied by a modification of anxiety-like behavior associated with endometriosis and effects on spontaneous pain were not evaluated in this work. Interestingly, THC produced opposite cognitive effects in sham and endometriosis mice. THC also induced an increase in markers of uterine innervation in sham animals, but prevented such changes in endometriosis mice, suggesting again different effects of THC under chronic inflammatory conditions. Importantly, THC also inhibited the growth of ectopic endometrium without apparent consequences on the eutopic endometrium and ovarian tissues. Altogether, the present data obtained in a preclinical model of endometriosis underline the interest in conducting clinical research to assess the effects of moderate doses of THC on endometriosis patients. Based on our results, we (clinicaltrials.gov, #NCT03875261) and others (gynica.com) have planned the initiation of clinical trials to provide evidence on the translatability of these results to women with endometriosis. These novel clinical trials will evaluate this new possible endometriosis treatment under pathological human conditions. However, cannabis has a large number of potential side effects, as well as a high potential for abuse liability (*Curran et al., 2016*), that have to be considered by physicians and patients. Therefore, the use of cannabis in unregulated scenarios should be discouraged taking into account these serious side effects.

# Materials and methods

## Key resources table

| Reagent type (species) or resource | Designation | Source or reference | Identifiers | Additional information |
|---|---|---|---|---|
| Strain, strain background (*Mus musculus,* female) | C57Bl/6J | Charles Rivers, Lyon, France | C57Bl/6J | Female |
| Chemical compound, drug | THC (Tetrahydrocannabinol) | THC-Pharm-GmbH | | Natural THC |
| Chemical compound, drug | Cremophor EL | Sigma-Aldrich | C5135; Kolliphor EL | |
| Chemical compound, drug | 0.9%, NaCl physiological saline | Laboratorios Ern | Vitulia | |
| Chemical compound, drug | Ethanol | Scharlab | ET00051000 | |
| Chemical compound, drug | Vaporised Isoflurane in oxygen | Virbac | Vetflurane | 4% V/V for induction; 2.5% V/V for maintenance |
| Chemical compound, drug | Optimal cutting temperature compound | Sakura finetek | 25608–930 | Item code 4583 |
| Biological sample (*Equus asinus*) | Normal donkey serum | Sigma-Aldrich | D9663-10ML | 3% in PBS with 0.3% Triton X-100 |
| Biological sample (*Capra aegagrus hircus*) | Normal goat serum | Vector lab | S-1000 | |
| Antibody | Rabbit polyclonal anti-beta-III tubulin antibody | Abcam | ab18207 | (1:2000) |

*Continued on next page*

*Continued*

| Reagent type (species) or resource | Designation | Source or reference | Identifiers | Additional information |
|---|---|---|---|---|
| Antibody | Donkey polyuclonal anti-rabbit Alexa Fluor A488 antibody | Thermo Fisher Scientific | A21206 | (1:1000) |
| Antibody | Goat polyclonal anti-rabbit Alexa Fluor A555 antibody | Abcam | ab150078 | (1:1000) |
| Chemical compound, drug | DAPI Fluoromount-G mounting media | SouthernBiotech | 0100–20 | |
| Chemical compound, drug | Calcium EDTA (Sodium calcium edetate) | Sigma-Aldrich | Sodium calcium edetate | |
| Commercial assay or kit | Enzyme-linked immunosorbent assay; ELISA | Calbiotech | ES180S-100 | |
| Software, algorithm | NIH Image J software | Wayne Rasband | | |
| Software, algorithm | GraphPad Prism 8 | GraphPad Software, Inc | | |
| Software, algorithm | IBM SPSS 23 software | IBM Corporation | | |
| Software, algorithm | Smart 3.0 videotracking software | Panlab | | |

## Animals

Female C57Bl/6J mice (Charles Rivers, Lyon, France) were used in all the experiments. Mice were 8 weeks old at the beginning of the experiments and were housed in cages of 4 to 5 mice with ad libitum access to water and food. The housing conditions were maintained at 21 ± 1°C and 55 ± 10% relative humidity in controlled light/dark cycle (light on between 8 AM and 8 PM). Animals were habituated to housing conditions and handled for 1 week before the start of the experiments. All animal procedures were conducted in accordance with standard ethical guidelines (European Communities Directive 2010/63/EU and NIH Guide for Care and Use of Laboratory Animals, 8[th] Edition) and approved by autonomic (Generalitat de Catalunya, Departament de Territori i Sostenibilitat) and local (Comitè Ètic d'Experimentació Animal, CEEA-PRBB) ethical committees. Mice were randomly assigned to treatment groups and all experiments were performed blinded for pharmacological and surgical conditions.

## Drugs

THC was purchased from THC-Pharm-GmbH (Frankfurt, Germany) as natural THC with 98.8% purity. This source of natural THC has been widely used in multiple research studies (*Busquets-Garcia et al., 2018*; *Busquets-Garcia et al., 2011*; *Cutando et al., 2013*; *Flores et al., 2014*; *Forsberg, 1970*; *Gunasekaran et al., 2009*; *Lopez-Rodriguez et al., 2014*; *Morrison et al., 2011*; *Puighermanal et al., 2013*). To corroborate the purity of the THC samples, High Performance Liquid Cromatography – Ultraviolet (HPLC-UV) was used for cannabinoid analysis and Gas Chromatography and Flame Ionization Detection (GC-FID) for terpenes (Canna Foundation, Paterna, Spain). These analyses revealed no detectable amounts of other cannabinoids or terpenes (Source Data Files 2, 3 and 4). THC was diluted in a vehicle composed of 2.5% ethanol, 5% Cremophor EL (C5135, Sigma-Aldrich St. Louis, MO, USA), and 92.5% saline, and was administered subcutaneously in a volume of 5 ml/kg.

## Estrous cycle determination

The phase of the estrous cycle was assessed by histological examination of cells extracted by vaginal lavage (*Byers et al., 2012*) the day of the surgeries and the day of euthanasia. Briefly, mice were gently restrained and 20 µl of saline were flushed 5 times into the vagina. The resulting fluid was

placed onto gelatinized slides, stained with methylene blue and observed at 40X magnification under a light microscope (DM6000 B, Leica Biosystems, Nussloch, Germany).

## Surgical induction of endometriosis

Endometriotic lesions were surgically-induced as previously described (*Somigliana et al., 1999*), with some modifications. Briefly, uterine horns from donor mice at diestrus were excised, opened longitudinally and biopsied into four pieces (2 × 2 mm). Recipient mice were anesthetized with vaporized isoflurane in oxygen (4% V/V for induction; 2.5% V/V for maintenance) and a midline incision of 1 cm was made to expose the abdominal compartment. Endometriosis mice had four uterine fragments sutured to the parietal peritoneum, whereas sham-operated mice received four similar-sized fragments of abdominal fat. Transplanted tissues and abdominal muscle and skin were stitched using 6–0 black silk (8065195601, Alcon Cusi S.A., Barcelona, Spain).

## Experimental protocols

The nociceptive, affective and cognitive manifestations associated with the presence of ectopic endometrium were determined in a first experiment. After the measurement of baseline mechanical sensitivity (day −1), endometriosis or sham surgery was performed (day 0), and nociceptive responses were assessed again 7, 14, 21 and 28 days after surgery. Anxiety-like behavior and cognitive performance were evaluated on days 23 and 27, respectively. At the end of the experimental sequence (day 32), mice were euthanized by cervical dislocation for sample collection.

A second experiment was conducted to investigate the presence of generalized nociceptive sensitization. Nociceptive responses to hind paw mechanical stimulation were assessed before (day −4) and 16 days after surgery. In parallel, mechanical sensitivity of the caudal abdominal area was evaluated on days −2 and 14 after surgery. An additional evaluation of nocifensive behaviors to abdominal mechanical stimulation was performed on day 14.

A third experiment was conducted to obtain the ED50 of acute THC administration for the alleviation of mechanical hypersensitivity. Endometriosis and sham mice were tested in the von Frey assay after administration of different doses of THC (1.25, 2, 2.5 and 5 mg/kg) or vehicle. Measurements were done 45 min after subcutaneous administration of THC or vehicle at time points in which endometriotic lesions and hypersensitivity in the caudal abdomen were fully developed (days 33–41).

The effects of chronic THC or vehicle were evaluated in endometriosis and sham mice in a fourth experiment. Chronic treatment with THC (2 mg/kg) or vehicle administered once a day (9 AM) started on day 1 after surgery and lasted until day 32. Behavioral measures were conducted as in the first experiment. Mice were tested on the nociceptive paradigm 45 min after drug or vehicle administration and on the anxiety-like and memory tests 6 hr after administration. Mice were euthanized on day 32 by cervical dislocation for sample collection.

A fifth experiment with 4 sets of mice was conducted to investigate THC tolerance development once the pain symptomatology was established. One of the groups underwent a sham surgery and the other three received endometrial implants. The sham group and one of the endometriosis groups received vehicle from day 1 to 16; one of the endometriosis groups received vehicle for 13 days and on day 15, and acute doses of THC (2 mg/kg) on days 14 and 16; the last endometriosis group received a repeated treatment with a daily administration of THC (2 mg/kg) from day 7 to 16. All mice were tested for mechanical sensitivity in the caudal abdominal area and the hind paw 45 min after drug or vehicle administration on days −2, 7 and 14 (caudal abdomen), and −4 and 16 (hind paw), respectively. The effects of THC on nocifensive behavior were measured on day 14.

## Nociceptive behavior

Mechanical sensitivity was quantified by measuring the responses to von Frey filament stimulation of the caudal abdominal area or the right hind paw. Von Frey filaments (1.65, 2.36, 2.44, 2.83, 3.22 and 3.61 corresponding to 0.008, 0.02, 0.04, 0.07, 0.16 and 0.4 g; Bioseb, Pinellas Park, FL, USA) were applied in increasing order of force, 10 times each, for 1–2 s, with an inter-stimulus interval of 5–10 s. Abrupt retraction of abdomen, immediate licking, jumping and scratching of the site of application were considered positive responses in the evaluation of abdominal mechanical sensitivity. Paw withdrawal, shaking or licking was considered a positive response in the evaluation of paw mechanical sensitivity. The area under the curve (AUC) was calculated by applying the linear trapezoidal rule to

the plots representing the frequency of response versus the numbers of von Frey filaments, which represent the logarithm of the filament force expressed in mg x 10.

## Nocifensive behavior

Unpleasantness of pain in response to a mechanical stimulus was measured as previously described (*Corder et al., 2019*; *Corder et al., 2017*) with minor modifications. Briefly, this parameter was evaluated using a single application of the von Frey filament 4.08 (corresponding to 1 g) against the caudal abdominal area shown in *Figure 1—figure supplement 1b*. The time spent protecting the area by guarding or seeking escape during the following 30 s was considered nocifensive behavior.

## Anxiety-like behavior

The elevated plus maze test was used to evaluate anxiety-like behavior in a black Plexiglas apparatus consisting of 4 arms (29 cm long x 5 cm wide), 2 open and 2 closed, set in cross from a neutral central square (5 × 5 cm) elevated 40 cm above the floor. Light intensity in the open and closed arms was 45 and 5 lux, respectively. Mice were placed in the central square facing one of the open arms and tested for 5 min. The percentages of time and entries to the open arms were determined as 100 x (time or entries to open arms) / (time or entries to open arms + time or entries to closed arms) as a measure of anxiety-like behavior.

## Cognitive behavior

The novel object recognition task was assayed in a V-shaped maze to measure cognitive performance (*Puighermanal et al., 2009*). On the first day, mice were habituated for 9 min to the maze. On the second day, mice were placed again in the maze for 9 min and two identical objects were presented at the ends of the arms of the maze. Twenty-four h later, one of the familiar objects was replaced with a novel one and mice were placed back in the maze for 9 min. The time spent exploring each object (novel and familiar) was recorded and a discrimination index (DI) was calculated as the difference between the time spent exploring the novel and the familiar object, divided by the total time exploring the two objects. A threshold of 10 s of total interaction with the objects was set to discard low levels of general activity.

## Sample harvesting and tissue preparation

Endometriotic lesions, uterine horns and ovaries were harvested from each mouse and fixed in 4% paraformaldehyde in phosphate buffered saline (PBS) for 4 hr and cryoprotected in 30% sucrose with 0.1% sodium azide for 6 days at 4°C. Samples were then embedded in molds filled with optimal cutting temperature compound (4583, Sakura Finetek Europe B.V., Alphen aan den Rijn, The Netherlands) and stored at −80°C until use.

## Histology and immunostaining

Endometriotic lesions and uteri were serially sectioned at 20 µm with a cryostat (CM3050, Leica Biosystems, Nussloch, Germany), mounted onto gelatinized slides and stored at −20°C until use. Sections of endometriotic lesions and uteri were stained with hematoxylin and eosin and observed under a Macro Zoom Fluorescence Microscope (MVX10, Olympus, Tokyo, Japan) for assessment of diameter and histological features.

Cyst sections were blocked and permeabilized with 3% normal donkey serum in PBS with 0.3% Triton X-100 for 2 hr and incubated overnight with rabbit anti-beta-III tubulin antibody (ab18207, 1:2000, Abcam, Cambridge, United Kingdom) in 3% normal donkey serum in PBS with 0.3% Triton X-100 at 4°C. After washing with PBS, sections were incubated for 1 hr at room temperature with anti-rabbit Alexa Fluor A488 antibody (A21206, 1:1000, Thermo Fisher Scientific, Waltham, MA, USA). Slides were washed with PBS and coverslipped with DAPI Fluoromount-G (0100–20, SouthernBiotech, Birmingham, AL, USA) mounting media.

Uterine sections were blocked and permeabilized with 5% normal goat serum in PBS with 0.3% Triton X-100 for 2 hr and incubated overnight with rabbit anti-beta-III tubulin antibody (ab18207, 1:2000, Abcam) in 5% normal goat serum in PBS with 0.3% Triton X-100 at 4°C. After washing with PBS, sections were incubated for 1 hr at room temperature with anti-rabbit Alexa Fluor A555

antibody (ab150078, 1:1000, Abcam, Cambridge, United Kingdom). Slides were washed with PBS and coverslipped with DAPI Fluoromount-G (0100–20, SouthernBiotech).

### Image analysis

Images of immunostained sections of cysts and uteri were captured with the X2 objective of a Macro Zoom Fluorescence Microscope (MVX10, Olympus, Shinjuku, Tokyo, Japan) and processed and quantified using the NIH Image J software. An observer who was blinded to treatment group assignment converted from 4 to 8 images per animal into negative black-and-white images and the threshold was manually adjusted. Images were then dilated, skeletonized and the mean percentage of immunoreactive area was obtained by running the 'Analyze particles' function.

### Determination of 17 β-estradiol plasma levels

Plasma samples were collected the day of euthanasia in tubes containing calcium EDTA. 17β-estradiol levels were determined with an enzyme-linked immunosorbent assay - ELISA (ES180S-100, Calbiotech, El Cajon, CA, USA) according to manufacturer instructions.

### Ovarian follicle counting

Sections of ovaries were stained with hematoxylin and eosin and observed under an upright microscope (DM6000 B, Leica Biosystems). The number of pre-antral and antral follicles was determined in every nine sections. Only follicles containing an oocyte were counted and the total number of follicles was estimated by multiplying the raw counts by nine according to published criteria (*Myers et al., 2004*). The number of corpora lutea was determined by direct counting of every 18 sections according to the average corpus luteum diameter (*Numazawa and Kawashima, 1982*).

### Statistical analysis

Data obtained with the nociception model were analyzed using one-way repeated measures ANOVA (surgery as between-subject factor), two-way repeated measures ANOVA (surgery and treatment as between-subject factors) or mixed models (surgery and treatment as between-subject factors and time as within-subject factor) whenever appropriate. Dose-response curve was fitted and ED50 determined using GraphPad Prism 8 (San Diego, CA, USA). Data obtained with the elevated plus maze test, novel object recognition task, histology, immunostaining and ovarian follicle counting were analyzed using a Student t-test (surgery) or a two-way ANOVA (surgery and treatment). Post hoc Bonferroni analysis was performed after ANOVA when appropriate. The nonparametric Kruskal-Wallis test was used whenever data did not have a normal distribution or equal variances, followed by Mann Whitney U when appropriate. Correlation between variables was determined using the Pearson correlation coefficient. Data are expressed as individual data points and mean ± SEM, and statistical analyses were performed using IBM SPSS 23 software (Chicago, IL, USA). The differences were considered statistically significant when the p value was below 0.05.

## Acknowledgements

The authors thank Mercè Vilaró Blay, Berta Güell Villena and Astoria Moores for their help and technical expertise.

## Additional information

### Funding

| Funder | Grant reference number | Author |
|---|---|---|
| Instituto de Salud Carlos III | RD16/0017/0020 | Rafael Maldonado |
| Ministerio de Ciencia, Innovación y Universidades | SAF2017-84060-R-AEI/FEDER-UE | Rafael Maldonado |
| Agència de Gestió d'Ajuts Universitaris i de Recerca | ICREA Academia 2015 | Rafael Maldonado |

| Agència de Gestió d'Ajuts Universitaris i de Recerca | 2019FI_B2_00111 | Alejandra Escudero-Lara |
| Agència de Gestió d'Ajuts Universitaris i de Recerca | 2017 SGR 669 | Rafael Maldonado |

The funders had no role in study design, data collection and interpretation, or the decision to submit the work for publication.

### Author contributions

Alejandra Escudero-Lara, Data curation, Formal analysis, Validation, Investigation, Visualization, Methodology, Writing - original draft, Conducted the behavioral and molecular experiments, Performed immunohistochemistry and microscopy, Wrote the manuscript; Josep Argerich, Formal analysis, Investigation, Methodology, Performed immunohistochemistry and microscopy; David Cabañero, Conceptualization, Supervision, Investigation, Visualization, Methodology, Writing - review and editing, Conceptualized and supervised the project, Participated in the experimental design, Wrote the manuscript; Rafael Maldonado, Conceptualization, Resources, Supervision, Funding acquisition, Investigation, Project administration, Writing - review and editing, Conceptualized, supervised and funded the project, Participated in the experimental design, Wrote the manuscript

### Author ORCIDs

Alejandra Escudero-Lara https://orcid.org/0000-0003-3728-2403
Josep Argerich https://orcid.org/0000-0003-4230-7089
David Cabañero https://orcid.org/0000-0002-1133-0908
Rafael Maldonado https://orcid.org/0000-0002-4359-8773

### Ethics

Animal experimentation: All animal procedures were conducted in accordance with standard ethical guidelines (European Communities Directive 2010/63/EU and NIH Guide for Care and Use of Laboratory Animals, 8th Edition) and approved by autonomic (Generalitat de Catalunya, Departament de Territori i Sostenibilitat) and local (Comitè Èticd'Experimentació Animal, CEEA-PRBB) ethical committees.

### Decision letter and Author response

Decision letter https://doi.org/10.7554/eLife.50356.sa1
Author response https://doi.org/10.7554/eLife.50356.sa2

## Additional files

### Supplementary files

• Transparent reporting form

### Data availability

All data supporting the findings of this study are available within the manuscript and its source data files. Source data files have been provided for Figures 1, 2, 3 and 4 and their figure supplements.

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
