## [Decision Letter]

**Acceptance summary:**

The Reviewers agree that since approved treatments for endometriosis target either reproductive hormones or prostaglandin synthesis, current medications do not address the anatomical cause of pain and have limited efficacy. Your report showing that THC simultaneously inhibits endometrial cysts and ameliorates mechanical hypersensitivity suggests THC may be a promising treatment option worthy of a clinical trial.

**Decision letter after peer review:**

Thank you for submitting your article "Disease-modifying effects of natural Δ9-tetrahydrocannabinol in endometriosis-associated pain" for consideration by *eLife*. Your article has been reviewed by three peer reviewers, and the evaluation has been overseen by a Reviewing Editor and Kate Wassum as the Senior Editor. The following individuals involved in review of your submission have agreed to reveal their identity: Rebecca Craft (Reviewer #1).

As there was more enthusiasm for the pain component of the study, we suggest that you focus your revision plan on the concerns related to this, as well as to the influence of the estrous cycle. There was more concern regarding the neuroinflammation and anxiety measures. One option is to consider removing these experiments and instead focus this report on pain. If you choose to keep these components in the report, please comment on how you will address each of the concerns raised by the reviewers on these aspects of the study.

Reviewer #1:

This manuscript describes studies undertaken to characterize the effects of THC in a mouse model of endometriosis. For the most part, the experiments are well-designed, the approaches/measures are appropriate, the results are presented clearly and succinctly, and the conclusions are logical. Overall this is a very interesting study with compelling results that should be of considerable interest to the cannabinoid scientific (and medical cannabis using and prescribing) communities. Addressing the issues noted below would further strengthen the manuscript.

1) Repeatedly administered THC was given starting on Day 1 (the day after surgical induction of endometriosis), but endometriosis symptoms as characterized in this model were shown to peak on Day 7. Thus, chronic THC may be attenuating the DEVELOPMENT of endometriosis than reducing established endometriosis. Additionally, the authors conclude that no tolerance develops to THC, but it appears they are basing this conclusion on the comparison of mice treated acutely with 2.0 mg/kg on Day 7 with those that were treated for 7 days leading up to Day 7 (i.e., during symptom development). A more appropriate test of tolerance (and a more valid model of human THC use for endometriosis) would be comparing acute THC on Day 14 in mice that had received vehicle for 13 days vs. those that had received THC daily starting on Day 7, when symptomatology is fully developed. These points should be noted in the discussion. THC may be much more effective at preventing development of pathology than reversing established pathology – which is what patients have.

2) The authors argue that the lack of effect of THC on the anxiety measure in contrast to its anti-allodynic effect reflects a dissociation between pain-induced anxiety and mechanical allodynia (or different potency of THC on these two endpoints, although that's implied rather than stated). What needs to be acknowledged as well is the possibility of different time courses of THC effect on allodynia vs. anxiety, since the former was assessed 45 min post-THC injection, and the latter was assessed 6 h post-THC injection. Do the authors have time course data using the s.c. route of THC administration for the plus-maze? Please revise the Discussion to acknowledge other possible interpretations.

3) No measures of spontaneous pain are shown, which is unfortunate because mechanical allodynia alone would seem to be a rather limited assessment of endometriosis pain. Can the authors use their plus maze data – specifically, total arm entries – to assess whether endometriosis suppresses activity in the maze, and whether THC reverses this effect? Otherwise, please acknowledge in the Discussion the potential limitation of measuring a single pain-related behavior, which may not translate well to the human experience of endometriosis pain.

Reviewer #2:

Escudero-Lara et al. uses a standard mouse model of endometriosis to study the effects of THC on pain behavior, learning, endometriotic growth/innervation, and hippocampal gene expression. This an important area of research since there is little other work on endometriosis, and it is essential to identify the therapeutic effects of molecules like THC. The study of multiple behavioral alterations in endometriosis with and without THC and even brain expression increases the potential widespread interest of their findings. Nevertheless, the inclusion of these additional modalities in this short-style paper gives short shrift to the explicit evaluation of pain other than von Frey responses.

The repetitive assessment of "responses" to von-Frey hairs throughout their protocol (to both acute with a dose-effect and chronic THC, endometriosis vs. sham) generates excellent confidence that their endometriosis model and reversal with THC has behavioral effects. However, there is a fundamental question whether this represents actual "pain behavior" or perhaps just anxiety at some experimenter poking near a prior incision site. The "pelvic" testing site is not clarified in the manuscript. Notably, the frequency of responsiveness is relatively consistent across a range force (Figure 1—figure supplement 1). This is not allodynia. Does this "hyperalgesia" generalize to other sites? It is important to establish the pelvic specificity of their effects and relevance to endometriosis. (Keep in mind the model does not study the three most problematic symptoms of endometriosis: dysmenorrhea, dyspareunia and infertility--This should be explicitly mentioned within the manuscript and is a major limitation). Therefore, it is absolutely critical to establish whether alterations in behavior in this model to a von Frey fiber reflect changes in pain sensitivity, responsiveness at a local or generalized level. Notably, studies for other pain models also use conditioned pain preference with an analgesic to establish levels of pain to verify pain behavior.

It is also imperative to verify that estrous cycle alterations do not contribute to potential differences between the group. It is possible that the authors already have data on estrous cycle effects (it was in their Materials and methods section), however the reviewer strongly encourages additional experiments confirming the model elicits endometriosis-like pain (location specific and indicative of actual pain not just responsiveness) which is reversed with THC.

In summary: If it were to be confirmed that endometriosis-related "pain" via a more specific assay and endometriosis cysts/ innervation were reduced accounting for any potential estrous cycle effects, this would be a remarkable paper. It is important to be particularly cautious as a high-impact publication of THC on a "model of endometriosis", will be considered strong evidence and encourage the use of cannabis in unregulated scenarios.

Reviewer #3:

This series of studies sought to determine if THC administration could ameliorate various pain-related, inflammatory and behavioral outcomes induced by experimental endometriosis. Endometriosis was experimentally induced by surgically implanting endometrium into the peritoneal wall of the pelvic cavity. This procedure resulted in pelvic hypersensitivity, anxiety and the generation of cognitive impairments, particularly in long term memory recall. Acute THC dosing was used to determine an effective dose of pain suppression and 2 mg/kg was sufficient to reduce mechanical hypersensitivity and was chosen for long term studies. Over 32 days of treatment, this dose of THC produced sustained alleviation of pain without tolerance and reversed memory impairments but had no impact on anxiety-like behavior. Endometriosis was also associated with elevated transcription of COX-2 and IL1b in the medial PFC, but not hippocampus, and this was also reversed by sustained THC administration. THC administration also reduced cyst size and uterine innervation. Overall, the data on THC effects on the endometriosis itself are interesting and compelling, but the behavioral and neuroinflammatory effects are moderate, at best, in magnitude and much less convincing relative to the tissue based changes.

The anxiety-like behavior analysis is incomplete, why is only closed arm entry data shown? Open arm entries and time as well as risk assessments, such as head dips, should also be displayed for the reader to be able to have a full understanding of the behavioral changes seen in this test as opposed to a selected outcomes. More so, this effect seems subtle at best given that it didn't even entirely replicate in the THC arm of the study where only% time, and not% entries, was found to still be elevated by endometriosis.

The neuroinflammatory changes are not very convincing. The effect seen from endometriosis alone is absent in Il1b and barely present in COX-2 expression, and THC does not have any impact on endometriosis itself as the THC endometriosis animals are virtually the same as the vehicle endometriosis animals, and these groups are not significantly different from one another. Almost all of these effects seem to be carried by the THC alone group, which is irrelevant for the endometriosis phenotype. The manner in which the manuscript is written is misleading as it makes it seem as if endometriosis alone produced a robust neuroinflammatory response that was reversed by THC, but inspection of the data clearly indicates this is not what was found. Also, the links made between this effect and the cognitive deficits seem highly improbable given the magnitude and lack of parallels in the directionality of the effects.

Could the authors provide the raw data on time spent interacting with the objects in the memory test, these effects can sometimes be very amplified by trivial differences in time interaction due to extremely low levels of interaction with the objects and this data will help to establish if this memory deficit is meaningful.

[Editors' note: further revisions were requested prior to acceptance, as described below.]

We very much appreciate the revisions to your manuscript. However, as you can see from the continuing comments of the Reviewers, further revisions are needed before the manuscript will be suitable for publication in *eLife*. We ask that you focus on your conclusions that are based on your most compelling results, which we believe are those that derive from the pain studies. It is also important to include caveats where the results are less convincing. In some areas, we strongly suggest that the studies be saved for a future report. All reviewers and editors agreed that with these revisions your manuscript will be very strong.

Eliminate the CPP data and related claims.

The reviewers did not find the CPP data to be convincing. They noted that you show mild CPA from THC in control conditions that is slightly muted in the endometriosis animals, largely because of a shift in baseline (the THC treatment groups in control and endometriosis are largely superimposable). This is difficult to interpret because of the alterations in baseline and the fact that the level of THC used is aversive. They also noted the lack of a dose-response curve for these data makes these data difficult to interpret.

Provide open arm time and entry data for the EPM data and temper conclusions regarding these data, being more honest about what these data are showing, which is a very mild anxiety phenotype.

The reviewers noted that open arm, not closed arm, measures are the typical convention in the field, and if not always representative of one another (if center time was also quantified, which is unclear). Additionally, the anxiety effect of endometriosis seems to be lost under conditions of repeated injections, making all of this data difficult to interpret.

As the dose of THC chosen appears to be anxiogenic in and of itself, please also highlight that since a dose-response curve was not done here.

The reviewers noted that you do not know if lower doses of THC would be able to impact this behavioral endpoint differently. Perhaps a lower dose of THC that would be ineffective for pain would be effective for anxiety.

Clarify that the primary tie to endometriosis is the anatomic similarly and abdominal sensitivity. Acknowledge that the evidence for spontaneous pain is weak and if it is present, it is not alleviated by THC, but that THC does reduce abdominal mechanical hypersensitivity and hyperalgesic behavior.

Acknowledge the limitations of this endometriosis model.

Clarify the interpretation of Figure 4B.

The reviewers noted that THC is associated with an increase in innervation in the sham, but not the endometriosis model.

Remove the display of coefficients in Figure 4.

---

## [Author Response]

Reviewer #1:This manuscript describes studies undertaken to characterize the effects of THC in a mouse model of endometriosis. For the most part, the experiments are well-designed, the approaches/measures are appropriate, the results are presented clearly and succinctly, and the conclusions are logical. Overall this is a very interesting study with compelling results that should be of considerable interest to the cannabinoid scientific (and medical cannabis using and prescribing) communities. Addressing the issues noted below would further strengthen the manuscript.1) Repeatedly administered THC was given starting on Day 1 (the day after surgical induction of endometriosis), but endometriosis symptoms as characterized in this model were shown to peak on Day 7. Thus, chronic THC may be attenuating the DEVELOPMENT of endometriosis than reducing established endometriosis. Additionally, the authors conclude that no tolerance develops to THC, but it appears they are basing this conclusion on the comparison of mice treated acutely with 2.0 mg/kg on Day 7 with those that were treated for 7 days leading up to Day 7 (i.e., during symptom development). A more appropriate test of tolerance (and a more valid model of human THC use for endometriosis) would be comparing acute THC on Day 14 in mice that had received vehicle for 13 days vs. those that had received THC daily starting on Day 7, when symptomatology is fully developed. These points should be noted in the Discussion. THC may be much more effective at preventing development of pathology than reversing established pathology – which is what patients have.

We conducted the proposed experiment to investigate the development of THC tolerance once symptomatology was fully established. Data (Results and Discussion section, subsection “Δ9-tetrahydrocannabinol alleviates pelvic pain, restores cognitive function and limits the growth of ectopic endometrium”, Figure 3B and Figure 3—figure supplement 2) showed that THC given for the first time on Day 14 was as effective as THC given on the same day after a daily treatment starting on Day 8. Therefore, we conclude that THC (2 mg/kg) does not lose its antinociceptive efficacy after repeated administration once pain symptomatology is established.

In addition, we highlighted in the Discussion that our experiments show an effect of chronic THC attenuating the development of endometriotic growths, but do not provide evidence of THC effects reducing established endometriotic lesions. The following sentence in the Results and Discussion section: “chronic THC showed an evident reduction in the size of endometrial cysts ….” was modified to “endometriosis mice receiving THC 2 mg/kg for 32 days showed an evident inhibition of the development of endometrial cysts” and the following sentence was added some lines below: “However, possible effects of THC on established endometriosis lesions were not evaluated.”

2) The authors argue that the lack of effect of THC on the anxiety measure in contrast to its anti-allodynic effect reflects a dissociation between pain-induced anxiety and mechanical allodynia (or different potency of THC on these two endpoints, although that's implied rather than stated). What needs to be acknowledged as well is the possibility of different time courses of THC effect on allodynia vs. anxiety, since the former was assessed 45 min post-THC injection, and the latter was assessed 6 h post-THC injection. Do the authors have time course data using the s.c. route of THC administration for the plus-maze? Please revise the Discussion to acknowledge other possible interpretations.

Previous work using the elevated plus maze in our laboratory has shown significant modifications of anxiety-like behavior 30 min and 4 h after acute THC exposure (Célérier et al., 2006; Puighermanal et al., 2013). In our present work, the test was performed 6 h after THC to evaluate possible impact of pain alleviation on anxiety-like behavior, but not to evaluate direct drug effects. Hence, we did not explore the time course of anxiety-like behavior after THC in the endometriosis model. To acknowledge different interpretations of the results on anxiety-like behavior the following sentences were added to the Results and Discussion section: “This evaluation was conducted 6 h after THC administration to study the impact of pain relief on anxiety-like behavior, rather than to assess direct drug effects. Thus, the absence of THC effects could indicate that pain alleviation is independent of anxiety-like responses and may also be associated to the different evaluation time points since the intrinsic effects of THC on anxiety have been revealed in time intervals shorter than 6 h (Célérier et al., 2006; Puighermanal et al., 2013)” and to the Conclusions section: “The pain-relieving effects of THC were not accompanied by a modification of anxiety-like behavior associated with endometriosis.”

3) No measures of spontaneous pain are shown, which is unfortunate because mechanical allodynia alone would seem to be a rather limited assessment of endometriosis pain. Can the authors use their plus maze data – specifically, total arm entries – to assess whether endometriosis suppresses activity in the maze, and whether THC reverses this effect? Otherwise, please acknowledge in the Discussion the potential limitation of measuring a single pain-related behavior, which may not translate well to the human experience of endometriosis pain.

We conducted a conditioned place preference experiment to investigate the efficacy of THC alleviating spontaneous pain. In a preconditioning session, sham and endometriosis mice were left in a box containing two main compartments separated by a triangular central division. Afterwards, the compartments were isolated, and mice received alternate injections of vehicle and THC paired to the different main compartments during a total of 8 conditioning sessions. In the test session, sham and endometriosis mice were left in the box with free access to the vehicle and the THC-paired compartments. The time spent in each compartment was compared to the time spent during the preconditioning session. Sham mice developed clear preference for the vehicle-paired compartment (p<0.01 vs. Preconditioning values), revealing an aversive effect of the THC treatment (2 mg/kg) in these experimental conditions. Interestingly, this effect was abolished in endometriosis mice (N.S. differences vs. Preconditioning values) suggesting beneficial effects of THC associated with spontaneous pain relief. As previously mentioned in the revision plan, aversive effects of the THC treatment could be expected, since we and others described aversive effects of THC depending on the dose and experimental conditions (Braida et al., Eur J Pharmacol 2004; Valjent and Maldonado, Psychopharmacology 2000; Lepore et al., Life Sci, 1995). The results of these experiments were added to Results and Discussion (subsection “Δ9-tetrahydrocannabinol alleviates pelvic pain, restores cognitive function and limits the growth of ectopic endometrium”, Figure 3D), and a new Conditioned place preference subsection has been added to Materials and methods. The conclusions have also been modified.

We also included a measure of pain unpleasantness (Corder et al., 2017; Corder et al., 2019) in response to a mechanical stimulus applied against the pelvic area, to provide an additional outcome of pain other than the increase in reflexive nociceptive responses. On day 14 after surgeries, a von Frey filament (4.08, corresponding to 1 g) was applied and the time spent protecting the area by guarding or seeking escape was measured for 30 s. Interestingly, untreated and vehicle-treated endometriosis mice showed a significant increase of the time displaying nocifensive behaviors when compared with untreated or vehicle-treated sham mice, respectively (Figures 1C and 3C). In this case, the subchronic treatment with THC (7 days of THC starting on day 8 after the surgery) was effective inhibiting this behavior, however the effects of acute THC administration given for the first time on day 14 were highly variable. This variable response could be associated to the aversive effects of the THC dose (2 mg(kg). Indeed, we have previously demonstrated that the first THC administration is particularly aversive and this behavioral response is modified after repeated THC exposure (Valjent and Maldonado, Psychopharmacology 2000). In agreement, THC treatments in unexperienced humans usually start with low doses and gradually increase until the effective dose is reached. Whenever adverse events are found, the patients return to the previous dose that was well tolerated. This procedure is followed in clinical settings (MacCallum and Russo, 2018), but could not be implemented in our model. Hence, it is likely that some of the exposed mice experienced aversive effects after the first THC exposure, in agreement with previous studies (Valjent and Maldonado, Psychopharmacology 2000). Methodology was included in the Materials and methods subsection Nocifensive behavior, and the Results and Discussion section was modified in subsection “Ectopic endometrium leads to pelvic pain sensitivity, anxiety-like behavior and memory impairment” and “Δ9-tetrahydrocannabinol alleviates pelvic pain, restores cognitive function and limits the growth of ectopic endometrium”.

We also assessed the effect of endometriosis and the THC treatment on the total arm entries of the plus maze and no significant differences between groups were found. These graphs were included in Figure 1D and Figure 3—figure supplement 4, and data were included in Source Data Files 1 and 3. Comments were added to subsection “Ectopic endometrium leads to pelvic pain sensitivity, anxiety-like behavior and memory impairment” and “Δ9-tetrahydrocannabinol alleviates pelvic pain, restores cognitive function and limits the growth of ectopic endometrium”. Absence of differences in total arm entries is an expected result since other mouse models of chronic pain lack alterations in this parameter (Benbouzid et al., Eur J Pain 2012; Chen et al., Neurosci Lett 2013; La Porta et al., 2015).

Reviewer #2:Escudero-Lara et al. uses a standard mouse model of endometriosis to study the effects of THC on pain behavior, learning, endometriotic growth/innervation, and hippocampal gene expression. This an important area of research since there is little other work on endometriosis, and it is essential to identify the therapeutic effects of molecules like THC. The study of multiple behavioral alterations in endometriosis with and without THC and even brain expression increases the potential widespread interest of their findings. Nevertheless, the inclusion of these additional modalities in this short-style paper gives short shrift to the explicit evaluation of pain other than von Frey responses.

We added new behavioral experiments to further characterize the pain component of endometriosis. The new data includes assessment of spontaneous pain with a conditioned place preference assay, evaluation of nociceptive sensitivity in distant areas, and measurement of pain unpleasantness through quantification of nocifensive behavior. The answers below describe the new results and the modifications included in the manuscript. Data on neuroinflammation will be saved for a future work following Editor suggestions.

The repetitive assessment of "responses" to von-Frey hairs throughout their protocol (to both acute with a dose-effect and chronic THC, endometriosis vs. sham) generates excellent confidence that their endometriosis model and reversal with THC has behavioral effects. However, there is a fundamental question whether this represents actual "pain behavior" or perhaps just anxiety at some experimenter poking near a prior incision site.

Both sham and endometriosis received an incision and showed increased response to von Frey filaments after one week. This increased sensitivity persisted in endometriosis mice, but sham-operated mice recovered baseline levels after two weeks. This reveals an effect of ectopic endometrium and not of incision itself.

The "pelvic" testing site is not clarified in the manuscript.

Mechanical sensitivity was quantified by measuring the responses to von Frey filament stimulation of the area cranial to the incision. A representation of the stimulated area was added to Figure 1—figure supplement 1b.

Notably, the frequency of responsiveness is relatively consistent across a range force (Figure 1—figure supplement 1). This is not allodynia.

A significant effect of the filament number was observed on days -1, 7, 14 and 21, indicating that the frequency of response to mechanical stimulation depends on the strength of the filament applied (Figure 1—figure supplement 1). The Author response table 1 shows significance levels of the effect of the filament number applied for each evaluation day.

**Author response table 1. resptable1:** Significance levels of the filament number in each evaluation day (Two-way repeated measures ANOVA).

	F	Sig.
Day -1	19.955	0.000
Day 7	4.655	0.001
Day 14	5.418	0.000
Day 21	6.670	0.000
Day 28	2.216	0.059

On day 28 after surgery, this effect became less evident because the stimulated area was more sensitive to stimulation with the thinnest filaments, indicating development of higher mechanical sensitivity to filaments that normally elicited a lower percentage of nociceptive responses. Data on the effect of filament number was added to Source Data Files 1, 2 and 3.

Does this "hyperalgesia" generalize to other sites? It is important to establish the pelvic specificity of their effects and relevance to endometriosis. (Keep in mind the model does not study the three most problematic symptoms of endometriosis: dysmenorrhea, dyspareunia and infertility. This should be explicitly mentioned within the manuscript and is a major limitation). Therefore, it is absolutely critical to establish whether alterations in behavior in this model to a von Frey fiber reflect changes in pain sensitivity, responsiveness at a local or generalized level. Notably, studies for other pain models also use conditioned pain preference with an analgesic to establish levels of pain to verify pain behavior.It is also imperative to verify that estrous cycle alterations do not contribute to potential differences between the group. It is possible that the authors already have data on estrous cycle effects (it was in their Materials and methods section), however the reviewer strongly encourages additional experiments confirming the model elicits endometriosis-like pain (location specific and indicative of actual pain not just responsiveness) which is reversed with THC.

As suggested by the reviewer, we complemented our data with measures of mechanical sensitivity in a site distant to the pelvis. Mechanical sensitivity in the hind paw was not affected by the presence of ectopic endometrium (Figure 1B). Paw nociception also remained unaltered after THC treatment in endometriosis and sham mice (Figure 3—figure supplement 3). Therefore, the mechanical hypersensitivity found in the pelvic area of endometriosis mice was not due to a generalized pain sensitization, but to local nociceptive responses in the pelvic zone. This information has been added to the Results and Discussion section, and the Materials and methods subsection Nociceptive behavior has been modified.

We conducted a conditioned place preference experiment to investigate the efficacy of THC alleviating spontaneous pain. In a preconditioning session, sham and endometriosis mice were left in a box containing two main compartments separated by a triangular central division. Afterwards, the compartments were isolated, and mice received alternate injections of vehicle and THC paired to the different main compartments during a total of 8 conditioning sessions. In the test session, sham and endometriosis mice were left in the box with free access to the vehicle and the THC-paired compartments. The time spent in each compartment was compared to the time spent during the preconditioning session. Sham mice developed clear preference for the vehicle-paired compartment (p<0.01 vs. Preconditioning values), revealing an aversive effect of the THC treatment (2 mg/kg) in these experimental conditions. Interestingly, this effect was abolished in endometriosis mice (N.S. differences vs. Preconditioning values) suggesting beneficial effects of THC associated with spontaneous pain relief. As previously mentioned in the revision plan, aversive effects of the THC treatment could be expected, since we and others described aversive effects of THC depending on the dose and experimental conditions (Braida et al., Eur J Pharmacol 2004; Valjent and Maldonado, Psychopharmacology 2000; Lepore et al., Life Sci, 1995). The results of these experiments were added to Results and Discussion (subsection “Δ9-tetrahydrocannabinol alleviates pelvic pain, restores cognitive function and limits the growth of ectopic endometrium”, Figure 3D), and a new Conditioned place preference subsection has been added to Materials and methods. The conclusions have also been modified.

We also included a measure of pain unpleasantness (Corder et al., 2017; Corder et al., 2019) in response to a mechanical stimulus applied against the pelvic area, to provide an additional outcome of pain other than the increase in reflexive nociceptive responses. On day 14 after surgeries, a von Frey filament (4.08, corresponding to 1 g) was applied and the time spent protecting the area by guarding or seeking escape was measured for 30 s. Interestingly, untreated and vehicle-treated endometriosis mice showed a significant increase of the time displaying nocifensive behaviors when compared with untreated or vehicle-treated sham mice, respectively (Figures 1C and 3C). In this case, the subchronic treatment with THC (7 days of THC starting on day 8 after the surgery) was effective inhibiting this behavior, however the effects of acute THC administration given for the first time on day 14 were highly variable. This variable response could be associated to the aversive effects of the THC dose (2 mg(kg). Indeed, we have previously demonstrated that the first THC administration is particularly aversive and this behavioral response is modified after repeated THC exposure (Valjent and Maldonado, Psychopharmacology 2000). In agreement, THC treatments in unexperienced humans usually start with low doses and gradually increase until the effective dose is reached. Whenever adverse events are found, the patients return to the previous dose that was well tolerated. This procedure is followed in clinical settings (MacCallum and Russo, 2018), but could not be implemented in our model. Hence, it is likely that some of the exposed mice experienced aversive effects after the first THC exposure, in agreement with previous studies (Valjent and Maldonado, Psychopharmacology 2000). Methodology was included in the Materials and methods subsection Nocifensive behavior, and the Results and Discussion section was modified.

(Keep in mind the model does not study the three most problematic symptoms of endometriosis: dysmenorrhea, dyspareunia and infertility--This should be explicitly mentioned within the manuscript and is a major limitation)

These caveats as well as limitations of our analysis of reproductive tissues are now acknowledged in the Results and Discussion section, as follows: “Numbers of preantral follicles, antral follicles and corpora lutea were similar in all groups in our experimental conditions (Figure 4—figure supplement 1C). These data suggest that endometriosis and THC were void of overt effects on ovarian follicle maturation and luteinization. However, other effects of endometriosis or THC on fertility cannot be excluded in our model. Similarly, the presence of prominent symptoms of endometriosis such as dysmenorrhea or dyspareunia could not be evaluated.”

It is also imperative to verify that estrous cycle alterations do not contribute to potential differences between the group. It is possible that the authors already have data on estrous cycle effects (it was in their Materials and methods section)In summary: If it were to be confirmed that endometriosis-related "pain" via a more specific assay and endometriosis cysts/ innervation were reduced accounting for any potential estrous cycle effects, this would be a remarkable paper. It is important to be particularly cautious as a high-impact publication of THC on a "model of endometriosis", will be considered strong evidence and encourage the use of cannabis in unregulated scenarios.

The phase of the estrous cycle was assessed the day of the surgeries and the day of euthanasia. This information has been added to Materials and methods section, Estrous cycle determination subsection. Estrous cycle was not assessed during the course of the experiment to avoid behavioral alterations due to stress-associated manipulation and vaginal stimulation.

In our experiments, we intentionally preserved the natural estrous cycle of mice without using artificial modifications such as hormonal supplementation with or without ovariectomization. Hormonal supplementation would certainly have altered our behavioral outcomes. As expected, there were variations in the estrous cycle of our mice at the end of the experimental sequence. Endometriosis and sham groups treated with vehicle or THC had similar distribution of mice in each cycle phase as shown in Author response table 2:

**Author response table 2. resptable2:** Number of animals in each phase of the cycle.

*Surgery*	*Treatment*	*Phase of the estrous cycle*	*N*
*Sham*	*Vehicle*	*Proestrus*	*0*
*Estrus*	*5*
*Metestrus*	*0*
*Diestrus*	*5*
*THC*	*Proestrus*	*1*
*Estrus*	*3*
*Metestrus*	*0*
*Diestrus*	*4*
*Endo*	*Vehicle*	*Proestrus*	*1*
*Estrus*	*2*
*Metestrus*	*0*
*Diestrus*	*7*
*THC*	*Proestrus*	*1*
*Estrus*	*3*
*Metestrus*	*0*
*Diestrus*	*6*

Despite the heterogeneity of the groups, differences in diameter and endometrial area of the lesions and uterine innervation were still evident as shown in Figures 4A and 4B. Author response image 1–Author response image 3 show the lack of effect of estrous cycle on cyst diameter, cyst endometrial area and uterine innervation.

**Author response image 1. respfig1:** Effects of THC and stage of the estrous cycle on cyst diameter. (**a**) Cyst diameter in the different group treatments and estrous cycle stages. A significant effect of the treatment was found. (**b**) Significance of the effects according to a Two-way ANOVA. Error bars are mean ± SEM. *p<0.05 vs vehicle. Endo, endometriosis.

**Author response image 2. respfig2:** Effects of THC and stage of the estrous cycle on cyst endometrial area. (**a**) Cyst endometrial area in the different group treatments and estrous cycle stages. (**b**) Shapiro-Wilk test revealed that data on endometrial area did not follow a normal distribution. Subsequent Kruskal-Wallis test showed no significant difference among these groups. Error bars are mean ± SEM. Endo, endometriosis.

**Author response image 3. respfig3:** Effects of THC and stage of the estrous cycle on uterine innervation. (**a**) Uterine innervation (represented as% of beta-III tubulin immunoreactive area) in the different experimental groups and estrous cycle stages. (**b**) Significance of the effects according to a Three-way ANOVA. A significant effect of the interaction between surgery and treatment was found as shown in Figure 4B. However, no effect of the estrous cycle was detected. Error bars are mean ± SEM. Endo, endometriosis.

The number of mice in each cycle stage and experimental condition may have been not high enough to estimate a possible estrogenic influence on our data (Power estimates of 0.172 for cyst size and 0.180 for uterine innervation). Therefore, we decided to analyze 17 β-estradiol plasma levels in the blood samples of mice from the chronic THC experiment (Figures 3-4). To quantify this parameter, we used an enzyme-linked immunosorbent assay (ES180S^-1^00, Calbiotech, El Cajon, CA, USA). The correlation between 17 β-estradiol levels and phase of estrous cycle, uterine innervation and lesion size was investigated. As expected, 17 β-estradiol plasma levels were dependent on the phase of the estrous cycle: mice in proestrus had the highest concentration (p<0.05 vs. diestrus and estrus) followed by mice in diestrus, and mice in estrus showed the lowest levels (Figure 4C, left panel). We found that levels of 17 β-estradiol were similar in all experimental groups (Figure 4C, right panel), although the estrogen levels were positively correlated with cyst diameter (Figure 4E, left), proving the estrogenic influence on ectopic endometrial lesions. Estrogen levels were not correlated with endometrial area of the cysts or uterine innervation (Figure 4E, middle and right panels), suggesting independent THC effects on these histological changes.

It is important to be particularly cautious as a high-impact publication of THC on a "model of endometriosis", will be considered strong evidence and encourage the use of cannabis in unregulated scenarios.

Risks of cannabis consumption are now highlighted in the Conclusions section: “However, cannabis has a large number of potential side effects, as well as a high potential for abuse liability (Curran et al., 2006), that have to be considered by physicians and patients. Therefore, the use of cannabis in unregulated scenarios should be discouraged taking into account these serious side effects”.

Reviewer #3:This series of studies sought to determine if THC administration could ameliorate various pain-related, inflammatory and behavioral outcomes induced by experimental endometriosis. Endometriosis was experimentally induced by surgically implanting endometrium into the peritoneal wall of the pelvic cavity. This procedure resulted in pelvic hypersensitivity, anxiety and the generation of cognitive impairments, particularly in long term memory recall. Acute THC dosing was used to determine an effective dose of pain suppression and 2 mg/kg was sufficient to reduce mechanical hypersensitivity and was chosen for long term studies. Over 32 days of treatment, this dose of THC produced sustained alleviation of pain without tolerance and reversed memory impairments but had no impact on anxiety-like behavior. Endometriosis was also associated with elevated transcription of COX-2 and IL1b in the medial PFC, but not hippocampus, and this was also reversed by sustained THC administration. THC administration also reduced cyst size and uterine innervation. Overall, the data on THC effects on the endometriosis itself are interesting and compelling, but the behavioral and neuroinflammatory effects are moderate, at best, in magnitude and much less convincing relative to the tissue based changes.The anxiety-like behavior analysis is incomplete, why is only closed arm entry data shown? Open arm entries and time as well as risk assessments, such as head dips, should also be displayed for the reader to be able to have a full understanding of the behavioral changes seen in this test as opposed to a selected outcomes. More so, this effect seems subtle at best given that it didn't even entirely replicate in the THC arm of the study where only% time, and not% entries, was found to still be elevated by endometriosis.

Raw data on entries and time spent in the arms of the elevated plus maze are now provided in Source Data Files 1 and 3, and total arm entries are included in Figures 1D and Figure 3 —figure supplement 4.

Daily manipulation for subcutaneous drug administration in the chronic THC experiment (Figure 3E) constitutes an additional stressor for the mice. This is reflected in the higher percentages of entries to closed arms observed in control animals treated with vehicle (Figure 3—figure supplement 4) when compared with untreated sham animals (Figure 1D). Unfortunately, the increased anxiety-like behavior hindered this endometriosis-associated phenotype. An explanation was added to the figure legend of Figure 3—figure supplement 4.

In spite of the technical complexity of the behavioral paradigm, the effect of endometriosis on the percentage of time spent in the closed arms was still significant (Figure 3E) and the experiment suggests absence of THC effects on anxiety-like behavior in these conditions.

The neuroinflammatory changes are not very convincing. The effect seen from endometriosis alone is absent in Il1b and barely present in COX-2 expression, and THC does not have any impact on endometriosis itself as the THC endometriosis animals are virtually the same as the vehicle endometriosis animals, and these groups are not significantly different from one another. Almost all of these effects seem to be carried by the THC alone group, which is irrelevant for the endometriosis phenotype. The manner in which the manuscript is written is misleading as it makes it seem as if endometriosis alone produced a robust neuroinflammatory response that was reversed by THC, but inspection of the data clearly indicates this is not what was found. Also, the links made between this effect and the cognitive deficits seem highly improbable given the magnitude and lack of parallels in the directionality of the effects.

Based on the Editor suggestions and the concerns of reviewer #3, all the data and discussion about neuroinflammation were removed from the manuscript.

Could the authors provide the raw data on time spent interacting with the objects in the memory test, these effects can sometimes be very amplified by trivial differences in time interaction due to extremely low levels of interaction with the objects and this data will help to establish if this memory deficit is meaningful.

As in previous studies (Busquets-Garcia et al., 2011; Navarro-Romero et al., Neurobiol Dis 2019), a threshold of 10 s of total interaction with the objects was set to discard low levels of general activity. This information was added to Materials and methods section, Cognitive behavior subsection. Thus, the time spent exploring the objects during the novel object recognition test was always higher than 10 s and we did not find effects of endometriosis or THC treatments on total exploration time, as shown in Author response image 4:

**Author response image 4. respfig4:** Total time exploring objects in the novel object recognition test. The total time exploring the objects was not affected by (a,b) endometriosis or (**b**) THC treatment. Error bars are mean ± SEM. Student t-test (**a**) and two-way ANOVA (**b**). Endo, endometriosis.

Raw data of the time exploring each individual object and the statistical analyses were included in Source data files 1 and 3.

[Editors' note: further revisions were requested prior to acceptance, as described below.]

We very much appreciate the revisions to your manuscript. However, as you can see from the continuing comments of the Reviewers, further revisions are needed before the manuscript will be suitable for publication in eLife. We ask that you focus on your conclusions that are based on your most compelling results, which we believe are those that derive from the pain studies. It is also important to include caveats where the results are less convincing. In some areas, we strongly suggest that the studies be saved for a future report. All reviewers and editors agreed that with these revisions your manuscript will be very strong.Eliminate the CPP data and related claims.The reviewers did not find the CPP data to be convincing. They noted that you show mild CPA from THC in control conditions that is slightly muted in the endometriosis animals, largely because of a shift in baseline (the THC treatment groups in control and endometriosis are largely superimposable). This is difficult to interpret because of the alterations in baseline and the fact that the level of THC used is aversive. They also noted the lack of a dose-response curve for these data makes these data difficult to interpret.

Following editors and reviewers recommendations, CPP data were removed from Figure 3 and Materials and methods and related claims were deleted from Abstract, Materials and methods, Results and Discussion and Conclusions section.

Provide open arm time and entry data for the EPM data and temper conclusions regarding these data, being more honest about what these data are showing, which is a very mild anxiety phenotype.The reviewers noted that open arm, not closed arm, measures are the typical convention in the field, and if not always representative of one another (if center time was also quantified, which is unclear). Additionally, the anxiety effect of endometriosis seems to be lost under conditions of repeated injections, making all of this data difficult to interpret.

Percentages of time and entries to the open arms of the elevated plus maze replaced the data on closed arms (Figure 1D, Figure 3D) in accordance with the editor’s comments. Percentage of entries to open arms and total arm entries were also added to Figure 3D to provide the same data as in Figure 1. These percentages are determined as 100 x (time or entries to open arms) / (time or entries to open arms + time or entries to closed arms). This information was added to Materials and methods section, Anxiety-like behavior subsection. In addition, the Abstract and the Results and Discussion section was modified to temper the conclusions regarding these data and to give more accurate description of the anxiety-like phenotype as suggested:

– Abstract: “female mice develop pelvic mechanical hypersensitivity in the caudal abdomen, mild anxiety-like behavior and substantial memory deficits”

– Results and Discussion: “endometriosis mice exhibited enhanced anxiety-like behavior reflected in lower percentages of time and entries to the open arms of the elevated plus maze”

– Results and Discussion: “Additional experiments were conducted to assess the effects of THC on the anxiety-like behavior induced by endometriosis pain (Figure 3D). As in previous experiments, endometriosis mice showed a lower percentage of time in the open arms of the elevated plus maze (Figure 3D), revealing increased anxiety-like behavior. However, the percentage of entries to the open arms was similar in endometriosis and sham mice. Therefore, the anxiogenic-like effect of the ectopic endometrium in these experimental conditions was mild and the present model was not optimal to reveal the emotional component of this painful situation.”

As the dose of THC chosen appears to be anxiogenic in and of itself, please also highlight that since a dose-response curve was not done here.The reviewers noted that you do not know if lower doses of THC would be able to impact this behavioral endpoint differently. Perhaps a lower dose of THC that would be ineffective for pain would be effective for anxiety.

The Discussion about the anxiogenic effects of THC has now been modified as follows:

“No significant effects of repeated THC 2 mg/kg were observed on the percentages of time and entries, although THC-treated mice showed a subtle increase in anxiety-like behavior (Figure 3D, percentage of time in open arms). Previous studies described anxiogenic-like effects of slightly higher doses (3 mg/kg) in naïve male mice (Viñals et al., 2015), and anxiolytic-like effects when using lower doses (0.3 mg/kg, Puighermanal et al., 2013; Viñals et al., 2015). Thus, possible effects of THC alleviating pain-related anxiety-like behavior in endometriosis mice could be hindered by intrinsic anxiogenic effects of this THC dose. Therefore, doses with less pain-relieving efficacy could potentially be effective promoting anxiolytic-like effects considering the intrinsic effects of THC on emotional-like behavior. Alternatively, the absence of clear effects of THC on anxiety-like behavior may be associated to the evaluation time point, which was 6 h after administration to study the impact of pain relief on anxiety-like behavior, rather than to assess direct drug effects. Total arm entries were similar among groups (Figure 3D).”

Clarify that the primary tie to endometriosis is the anatomic similarly and abdominal sensitivity. Acknowledge that the evidence for spontaneous pain is weak and if it is present, it is not alleviated by THC, but that THC does reduce abdominal mechanical hypersensitivity and hyperalgesic behavior.

The manuscript was modified to emphasize that the mouse model mainly reproduces the histopathological features of endometriosis and the related abdominal hypersensitivity. Thus, the new version acknowledges a lack of evidence for THC alleviating spontaneous pain and highlights its effects reducing hypersensitivity of the caudal abdomen and the cognitive impairment associated with endometriosis. The terms “pelvic” or “pelvis” were changed to “abdominal” or “caudal abdomen / caudal abdominal area” thorough the manuscript to increase the accuracy of the terminology:

– Abstract: “In this model, female mice develop mechanical hypersensitivity in the caudal abdomen, mild anxiety-like behavior and substantial memory deficits associated with the presence of extrauterine endometrial cysts. Interestingly, daily treatments with THC (2 mg/kg) alleviate mechanical hypersensitivity and pain unpleasantness, modify uterine innervation and restore cognitive function without altering the anxiogenic phenotype”.

– Introduction section: “This work investigates the effects of natural THC in a mouse model of endometriosis that reproduces the ectopic endometrial growths and some of the behavioral alterations of clinical endometriosis. Our data show that THC is effective inhibiting hypersensitivity in the caudal abdominal area without inducing tolerance, as well as reducing the pain unpleasantness associated with endometriosis. Notably, THC also prevents the cognitive impairment observed in mice with ectopic endometrium without modifying anxiety-like behavior at this particular dose. Interestingly, THC shows efficacy limiting the development of ectopic endometrium, revealing disease-modifying effects of this natural cannabinoid”.

– Results and Discussion section: “Hence, mice with ectopic endometrium recapitulate in our model some of the symptomatology observed in the clinics, although manifestations of spontaneous pain could not be evaluated in this work”.

– Results and Discussion section: “Interestingly, we also found increased expression of the neuronal marker β-III tubulin in the uteri of endometriosis mice (Figure 1—figure supplement 3), mimicking not only some of the symptoms but also the histological phenotype observed in women with endometriosis (Miller EJ, 2015; Tokushige et al., 2006)”.

– Conclusions section: “chronic administration of a moderate dose of the phytocannabinoid THC relieves mechanical hypersensitivity of caudal abdominal area, pain unpleasantness and cognitive impairment associated with the presence of ectopic endometrial cysts. These behavioral manifestations correlate with a decrease in the size of ectopic endometrium in THC-exposed mice. However, the pain-relieving effects of this particular dose of THC were not accompanied by a modification of anxiety-like behavior associated with endometriosis and effects on spontaneous pain were not evaluated in this work”.

Acknowledge the limitations of this endometriosis model.

As indicated above, Introduction, Results and Discussion and Conclusions sections were modified to acknowledge that the model reproduces some of the symptoms observed in women with endometriosis, but the presence of spontaneous pain and other symptoms of endometriosis could not be revealed in the present work.

Clarify the interpretation of Figure 4B.The reviewers noted that THC is associated with an increase in innervation in the sham, but not the endometriosis model.

The distinct effects of THC on uterine innervation were discussed in the Results and Discussion section as follows:

“Repeated THC increased the expression of neuronal markers in the uteri of sham mice, similar to the increase provoked by the ectopic endometrium (Figure 4B). Interestingly, THC prevented this increase in endometriosis mice (Figure 4B) indicating again that THC exposure may have different consequences under chronic inflammatory conditions. In agreement, recent studies showed differential effects of THC on the nervous system of rodents with and without chronic inflammation (Bilkei-Gorzo et al., 2017; Sarne et al., 2017)”.

Remove the display of coefficients in Figure 4.

Coefficients were removed from Figure 4D and placed in the figure legend.